# Numerical Investigation of Scour Beneath Pipelines Subjected to an Oscillatory Flow Condition

Jun Huang [1], Guang Yin [2,*], Muk Chen Ong [2], Dag Myrhaug [3] and Xu Jia [1]

1   Department of Engineering Research and Design, CNOOC Research Institute, Beijing 100028, China;
    huangjun1@cnooc.com.cn (J.H.); jiaxu@cnooc.com.cn (X.J.)
2   Department of Mechanical and Structural Engineering and Materials Science, University of Stavanger,
    4036 Stavanger, Norway; muk.c.ong@uis.no
3   Department of Marine Technology, Norwegian University of Science and Technology, 7491 Trondheim,
    Norway; dag.myrhaug@ntnu.no
*   Correspondence: guang.yin@uis.no

**Abstract:** The present study carries out two-dimensional numerical simulations to investigate scour beneath a single pipeline and piggyback pipelines subjected to an oscillatory flow condition at a Keulegan–Carpenter (*KC*) number of 11 using SedFoam (an open-source, multi-dimensional Eulerian two-phase solver for sediment transport based on OpenFOAM). The turbulence flow is resolved using the two-phase modified $k - \omega$ 2006 model. The particle stresses due to the binary collisions and enduring contacts among the sediments are modeled using the rheology model of granular flow. The present numerical model is validated for the scour beneath a single pipeline, and the simulated sediment profiles are compared with published experimental data and numerical simulation results. The scour process beneath three different piggyback pipelines under the same flow condition are also considered, and the scour development and surrounding flow patterns are discussed in detail. Typical steady-streaming structures around the pipeline due to the oscillatory flow condition are captured. The scour depth during the initial development of the scour process for the piggyback pipeline with the small pipeline placed above the large one is the largest among all the investigated configurations. The phase-averaged flow fields show that the flow patterns are influenced by the additional small pipeline.

**Keywords:** scour; two-phase solver; SedFoam; piggyback pipelines

## 1. Introduction

Offshore pipelines are commonly used to transport oil or gas in industries. When these pipelines are placed on the sandy seabed, the local scour will be induced by the surrounding complex flow. Scour holes can be formed beneath the pipelines. Then, the pipelines become unsupported over the seabed and after the depth of the scour hole is deepened to an extent, the pipelines might undergo vortex-induced vibrations, which will influence the on-bottom stability and fatigue life of the pipelines. Therefore, the prediction of the interaction between the fluid flow, the pipelines and the sediment transport near the eroded seabed is essential. There have been extensive studies both by experiments (Mao [1]; Sumer and Fredsøe [2]) and numerical simulations (Fuhrman et al. [3]; Mathieu et al. [4]) carried out to investigate scour beneath offshore pipelines. It was found that the general process of scour includes the substage of the onset, tunnel and lee-wake erosion. The onset of scour is caused by seepage flow within the sediment layer beneath the pipeline induced by the hydraulic pressure difference between the upstream and downstream sides of the pipeline. Then, the scour process undergoes tunnel erosion characterized by a small gap formed between the pipeline and the seabed after the mixture of the sediment and water. The gap is then enlarged by the sediment transport due to the strong shear stress caused by the high-speed flow through the gap. Finally, the lee-wake stage appears when there is downstream

convection of the sand dune behind the pipeline caused by the vortex shedding. On the seabed in deep water, the scour is usually caused by currents, while in shallow water regions, the pipelines can be subjected to a wave-induced oscillatory flow, where a more complex scour process beneath the pipeline will happen. It was found by Sumer and Fredsøe [2] using experimental measurement that the equilibrium scour depth mainly depends on the Keulegan–Carpenter number (*KC*) defined as $KC = U_\mathrm{m} T_w / D$, where $U_\mathrm{m}$ is the amplitude of the undisturbed near-bed orbital velocity, $T_w$ is the wave period and $D$ is the diameter of the pipeline, instead of the Shields parameter. The relationship between the equilibrium scour depth and *KC* is summarized using regression. In addition, there is backfilling of the scour hole under an ever-changing wave and current climate according to Sumer et al. [5].

Apart from experimental studies carried out by Sumer and Fredsøe [2], Fredsøe et al. [6], Cheng et al. [7] and Qi and Gao [8], the scour process under waves at different *KC*s was also investigated using a stochastic method by Myrhaug et al. [9] and numerical simulations by Liang and Cheng [10], Kazeminezhad et al. [11], Fuhrman et al. [3] and Li et al. [12]. The main challenge in numerical simulations of the scour process is how to resolve the interactions among the flow, the sediments, and the structures. The present numerical tools in resolving the fluid and sediments can be categorized into single-phase and two-phase models. For the single-phase models, which were widely used by Liang and Cheng [10], Liu and Garcia [13], Fuhrman et al. [3], Zhao et al. [14], Baykal et al. [15] and Li et al. [12], a morphology model of the seabed is applied with a bed-load transport model and a suspended-load transport model. The bed-load transport is usually represented by semi-empirical formulations. For the two-phase models developed recently by Hsu and Liu [16], Lee et al. [17], Cheng et al. [18] and Mathieu et al. [4], the transport of the sediment is also governed by mass and momentum conservation equations similar to the fluid flow and no bed-load transport model is assumed. Compared with the single-phase models, the complicated physical phenomenon of sediment transport can be resolved more completely. However, the current applications of the two-phase model to investigate the scour process are rare. In the present study, a recently developed open-source Eulerian two-phase solver based on OpenFOAM (Cheng et al. [18]; Chauchat et al. [19]) is used to study the local scour under an oscillatory flow condition.

In most of the previous studies, the scour beneath a single pipeline, two pipelines (Zhao et al. [14]; Zhang et al. [20]; Li et al. [12]) or around piles (Baykal et al. [15]) under currents or waves was considered. For decades, a configuration called piggyback pipeline has been widely used, which consists of one primary large pipeline used to transport oil or gas and a small pipeline rigidly installed with the large one to transport chemicals. It was reported in Yang et al. [21] and Serta et al. [22] that hydrodynamic interference can be triggered by the additional small pipeline. Furthermore, the studies carried out by Zhao and Cheng [23], Zhao et al. [14], Zhao et al. [24] and Yang et al. [25] reported changes in the scour beneath a piggyback pipeline compared with that beneath a single pipeline under currents. However, there are no detailed studies on the oscillatory flow-induced scour beneath a piggyback pipeline. The objective of the present study is to evaluate how the presence of the small pipeline and different configurations of the piggyback pipeline influence the scour under a sinusoidal oscillatory flow condition. The surrounding flow patterns are also discussed. CFD simulations are carried out based on a two-phase solver. The paper is organized as follows. Section 2 gives a brief introduction on the numerical model in the present study. Mesh convergence studies and validation studies are conducted in Section 3. The results of the scour beneath the piggyback pipelines with three different relative locations between the large pipeline and the small pipeline are presented in Section 4. Finally, the conclusions are made in Section 5.

## 2. Mathematical Formulation

In the present study, a two-dimensional (2D), two-phase flow model is used, where both the sediment and fluid phases are assumed to be governed by the mass conservation

and momentum conservation equations. The mass conservation equations for both the two phases are given as:

$$\frac{\partial \phi}{\partial t} + \frac{\partial \phi u_i^s}{\partial x_i} = 0 \tag{1}$$

$$\frac{\partial (1-\phi)}{\partial t} + \frac{\partial (1-\phi) u_i^f}{\partial x_i} = 0 \tag{2}$$

where $\phi$ is the sediment concentration ($0 \le \phi \le 1$) within a cell: for $\phi = 0$, the cell is occupied by the fluid phase while for $\phi = 1$, the cell is occupied by the sediment phase. The quantities $u_i^s$ and $u_i^f$ are the mean velocities of the sediment (denoted by the superscript 's') and fluid phase (denoted by the superscript 'f'), respectively. $i = 1, 2$ denotes the streamwise and cross-stream direction, which are also denoted as $x, y$ in the present paper. The momentum conservation equations for the two phases are given by:

$$\frac{\partial \rho^s \phi u_i^s}{\partial t} + \frac{\partial \rho^s \phi u_i^s u_j^s}{\partial x_j} = -\frac{\partial p^s}{\partial x_i} - \frac{\partial \phi p^f}{\partial x_i} + \frac{\partial \tau_{ij}^s}{\partial x_j} + \rho^s \phi g \delta_{i2} + M_i^{sf} \tag{3}$$

$$\frac{\partial \rho^f (1-\phi) u_i^f}{\partial t} + \frac{\partial \rho^f (1-\phi) u_i^f u_j^f}{\partial x_j} = -\frac{\partial (1-\phi) p^f}{\partial x_i} + \frac{\partial \tau_{ij}^f}{\partial x_j} + \rho^f (1-\phi) g \delta_{i2} + M_i^{fs} \tag{4}$$

where $\rho^s$ and $\rho^f$ are the densities of the sediment and fluid phases, respectively. In the present study, $\rho^s = 2.6 \times 10^3$ kg/m$^3$ and $\rho^f = 1.0 \times 10^3$ kg/m$^3$ are used. $g = 9.8$ m/s$^2$ is the gravitational acceleration. $p^f$ is the fluid phase pressure and $p^s$ is the sediment-phase normal stress. $\tau_{ij}^s$ and $\tau_{ij}^f$ are the sediment phase and fluid phase shear stresses. The fluid shear stress is given as

$$\tau_{ij}^f = \rho^f (1-\phi) \left[ \nu_{Eff} \left( \frac{\partial u_i^f}{\partial x_j} + \frac{\partial u_j^f}{\partial x_i} - \frac{2}{3} \frac{\partial u_k^f}{\partial x_k} \delta_{ij} \right) - \frac{2}{3} k^f \delta_{ij} \right] \tag{5}$$

where $k^f$ is the turbulent kinetic energy of the fluid phase and $\nu_{Eff}$ is the effective viscosity given by $\nu_{Eff} = \nu_t + \nu^f$ with the fluid kinetic viscosity $\nu^f$ and the turbulent eddy viscosity $\nu_t$. The value of $\nu^f$ is set to be $1.0 \times 10^{-6}$ m$^2$/s for the water. In the present study, the turbulent eddy viscosity is resolved by using the two-phase $k - \omega$ 2006 turbulence model developed by Mathieu et al. [4] based on the revisited $k - \omega$ turbulence model by Wilcox [26]. The $k - \omega$ 2006 turbulence model was developed with an improvement on the $k - \omega$ SST model (Menter [27]) by modifying the additional cross diffusion term in the transport equation of $\omega$. According to Mathieu et al. [4], the $k - \omega$ 2006 turbulence model can reproduce the vortex-shedding phenomenon and qualitatively predict the lee-wake scour behind the pipeline compared with the results using the $k - \varepsilon$ turbulence model. The detailed formulations of the two transport equations can be found in Mathieu et al. [4].

The sediment phase normal stress $p^s$ and shear stress $\tau_{ij}^s$ in Equation (3) are both comprised of the stress due to the friction (denoted by the superscript 'f') and the stress induced by collision between the particles (denoted by the superscript 'c') as:

$$p^s = p^{sc} + p^{sf} \tag{6}$$

$$\tau_{ij}^s = \tau_{ij}^{sc} + \tau_{ij}^{sf} \tag{7}$$

In the present study, the rheology model of granular flow is adopted to determine these terms. A detailed description of the rheology model can be found in Mathieu et al. [4].

The last two terms $M_i^{sf} = -M_i^{fs}$ in Equations (3) and (4) represent the inter-phase momentum transfer, which takes the form of

$$M_i^{sf} = -M_i^{fs} = -\phi\beta\left(u_i^f - u_i^s\right) + \beta\frac{\nu_t}{\sigma_c}\frac{\partial\phi}{\partial x_i} + p^f\frac{\partial(1-\phi)}{\partial x_i} \tag{8}$$

The first two terms are associated with the drag force between the two phases, which is associated with the relative velocity difference between the fluid and the particles through the parameter $\beta$ given by the following formula:

$$\beta = \begin{cases} \frac{150\phi\nu^f\rho^f}{(1-\phi)d_{50}^2} + \frac{1.75\rho^f\left|\mathbf{u}^f - \mathbf{u}^s\right|}{d_{50}}, & \phi \geq 0.2 \\ \frac{0.75C_d\rho^f\left|\mathbf{u}^f - \mathbf{u}^s\right|(1-\phi)^{-1.65}}{d_{50}}, & \phi < 0.2 \end{cases} \tag{9}$$

where $d_{50}$ is the value of the medium diameter of the sediment and the drag coefficients $C_d$ can be determined based on the particle Reynolds number $Re_p = (1-\phi)\left|\mathbf{u}^f - \mathbf{u}^s\right|d_{50}/\nu^f$ as

$$C_d = \begin{cases} \frac{24\left(1+0.15Re_p^2\right)}{Re_p}, & Re_p \leq 1000 \\ 0.44, & Re_p > 1000 \end{cases} \tag{10}$$

The last two terms in (9) represent the fluid suspension and the inter-phase pressure correction, respectively.

## 3. Computational Setup

The computational domain for the present study is shown in Figure 1a. The length of the computational domain is set to be $66D$ and the height is set to be $13D$, where $D = 0.03$ m is the diameter of the pipeline according to the model test reported by Sumer and Fredsøe [2], Fuhrman et al. [3] and Zhao et al. [28]. The distance between the inlet boundary and the center of the large pipeline is $Lu = 33D$, half of the total length of the computational domain. The distance between the top boundary and the center of the large pipeline is $10D$ and the height of the sediment layer is set to be $2.5D$. The chosen computational domain size is even larger than that reported in Liang and Cheng [10] (where the length is $60D$ and the height is $10D$) to suppress the far field effects of applying an oscillatory flow condition at the inlet boundary. For the simulations of the scour beneath a piggyback pipeline, a smaller pipeline with a diameter of $d = D/3$ is placed aside the main pipeline with a gap ratio of $e = 0.25D$. This piggyback configuration as shown in Figure 1b is similar to that reported in Zhao et al. [24]. Three relative angles of $\alpha = 90°, 45°, 0°$ between the two pipelines are considered.

The boundary conditions for the numerical simulations are listed as follows:

- At the surfaces of the pipelines and at the bottom boundary of the computational domain, a no-slip boundary condition is applied for velocities of the two phases. A zero gradient boundary condition is applied for the sediment concentration $\phi$. For $k^f$ and $\omega^f$, the standard wall-function boundary conditions are used.
- At the top boundary, the velocities of the two phases, the sediment concentration $\phi$, $k^f$ and $\omega^f$ are assumed to be zero normal gradient.
- At the inlet, the following boundary conditions are used, which are similar to those used in Fuhrman et al. [3]:

$$u^s = u^f = U_{\mathrm{m}}\sin\left(\frac{2\pi t}{T_w}\right), v^s = v^f = 0 \tag{11}$$

$$k^f = k_{\mathrm{m}}\left(\sin\left(\frac{2\pi t}{T_w}\right)\right)^2, k_{\mathrm{m}} = 0.0005U_m^2 \tag{12}$$

$$\omega^f = \omega_{\mathrm{m}}\left|\sin\left(\frac{2\pi t}{T_w}\right)\right|, \omega_{\mathrm{m}} = \frac{k_m}{100\nu^f} \tag{13}$$

In these wave-form formulas, $U_m$ is the amplitude of the undisturbed near-bed orbital velocity and $T_w$ is the period. The $KC$ defined using the two parameters is set to be 11 for comparison with the experimental data reported by Sumer and Fredsøe [2] in the present study. The values of Reynolds number based on $U_m$ and the diameters of the large and small pipelines are 7200 and 2400, respectively. A zero gradient boundary condition is applied for the pressures of the two phases. A smooth sediment concentration profile at the inlet is defined using an approximate analytical expression of $\phi(y) = \phi_0/2 + \phi_{m0}(1 + \tan h((-y_b - y)/A))$, which can provide a relatively smooth transition from $\phi_{m0}$ at the bed to $\phi_0 \approx 0$ in the upper fluid domain, as suggested by Cheng [29]. The constants of $\phi_{m0} = 0.6128$ and $A = 5 \times 10^{-4}$ are used in the present study and the value of $y_b$ is selected such that the value of $\phi$ at $y = -D/2$ is close to 0. The profile of $\phi(y)$ is shown in Figure 1c. This analytical expression and the constant values are similar to those used in Mathieu et al. [4]. The inlet value for $\phi(y)$ is also used for the initial value of the sediment concentration in the computation domain, and $1 - \phi(y)$ is used for the initial value of the fluid concentration.

- At the outlet boundary, zero-gradient boundary conditions are imposed for the velocities of the two phases and the values of $k^f$ and $\omega^f$. A hydrostatic pressure is used for the fluid pressure $p^f$.

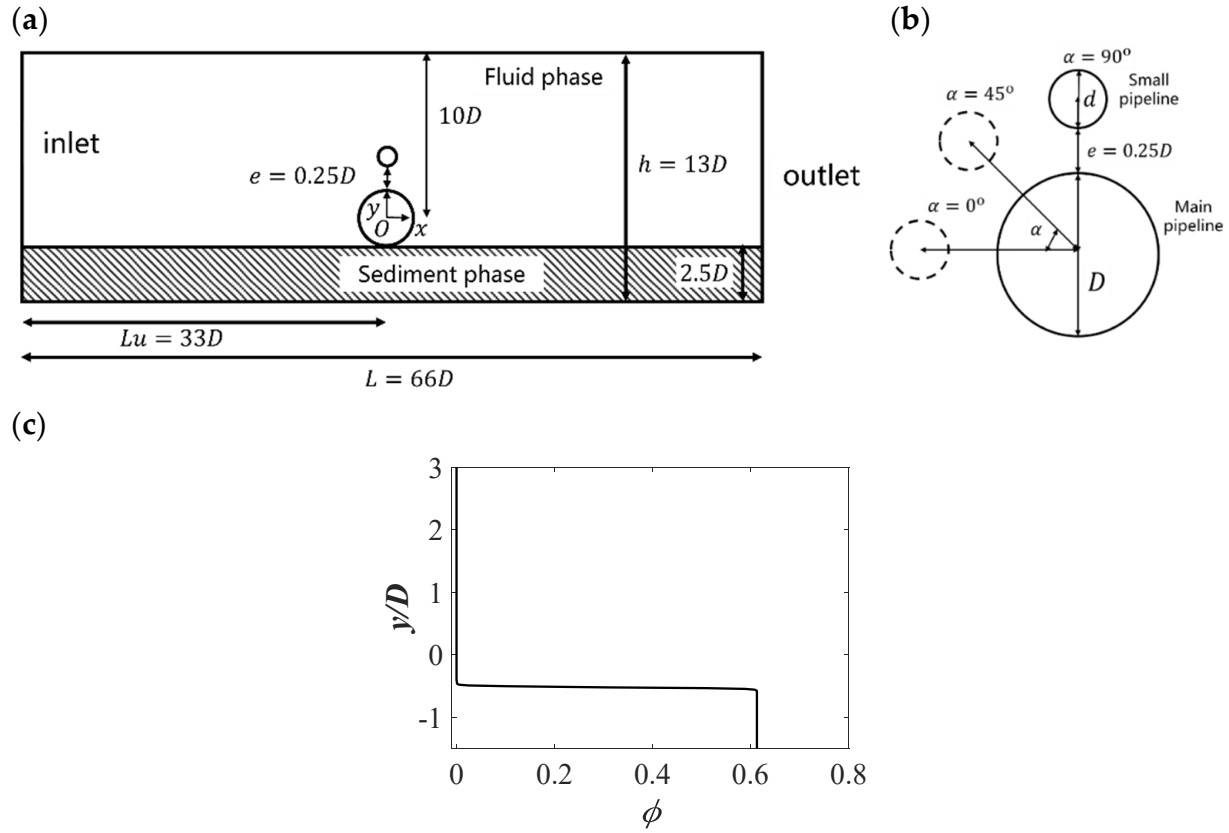

**Figure 1.** (**a**) Computational domain; (**b**) the configurations of the piggyback pipelines within the computational domain; (**c**) the profile of the sediment concentration $\phi(y)$.

The mesh convergence studies are carried out for the single pipeline case to determine the optimal grid resolutions. For all the meshes in the convergence study, a structural mesh is first built for the rectangular computational domain based on the mesh numbers in the $x$ and $y$ directions shown in Table 1. Then, the grids are progressively refined in the region around the pipeline and the interface between the two phases, as seen in Figure 4. The grid size around the interface ranges from $5 \times 10^{-4}$ to $7.5 \times 10^{-4}$ according to Mathieu et al. [4]. The expansion ratio of the grids near the surface of the pipeline is kept as 1.2. For all the

simulations, the time steps are chosen to maintain the maximum Courant–Friedrichs–Lewy (CFL) number as less than 0.5. The residual values for the pressure and velocities of two phases are kept below $10^{-6}$ and the residual value for $\phi$ is kept below $10^{-9}$ after interactions at every time step for all simulations. The mesh is refined around the surface of the cylinder and around the interface between the fluid and sediment phases. Three different meshes from coarse to fine for Meshes 1 to 3 are chosen, and their grid numbers are presented in Table 1. First, the sediment profiles obtained using the three mesh resolutions at three selected time instants are shown in Figure 2 to evaluate the effects of the grid resolution on the scour prediction. The sediment profiles are denoted as the iso-surface corresponding to $\phi = 0.5$, according to Mathieu et al. [4]. It can be seen that there is discrepancy between the ripples along the flat part of the sediment surface and the shapes of the scour holes around the outer part of the scour holes, which may be due to the sensitivity of the captured sediment profiles arising from the unsteadiness of the oscillatory flow condition. However, the overall shapes of the scour hole beneath the pipeline are similar for different meshes and the shapes of the scour holes around the parts with the largest scour depths appear similar for Meshes 2 and 3. Then, the drag and lift coefficients of the cylinder are used to evaluate the effects of grid convergence on the predictions of the hydrodynamic quantities. The values of the coefficients are calculated as

$$C_d = \frac{F_x}{0.5\rho^f DU_m^2} \tag{14}$$

$$C_l = \frac{F_y}{0.5\rho^f DU_m^2} \tag{15}$$

where $F_x$ and $F_y$ are the forces on the cylinder acted by the fluid phase in the horizontal and the cross-section directions, respectively. The time histories of $C_d$ and $C_l$ after the flow states become steady are shown in Figure 3. It can be seen that the drag coefficients obtained using the three mesh resolutions are close to each other. There is a difference of the predicted lift forces between cases, which may be due to the sensitivity of the lift force under the unsteadiness of the oscillatory flow. The root-mean-squares (rms) of $C_d$ and $C_l$ are shown in Table 1. It is worth mentioning that due to the symmetry of the oscillatory flow, the time-averaged values of $C_d$ should be zero and are not shown here. The relative difference of the root-mean-squares between different cases is less than 5%. Therefore, based on the above results, it can be concluded that the grid resolutions of Mesh 2 can be regarded as sufficient to predict the scour hole as well as the hydrodynamic quantities and are then used for the piggyback pipeline cases. An example of Mesh 2 is shown in Figure 4 to give an overview of the mesh density.

**Table 1.** Comparisons of hydrodynamic forces on the single pipeline for different mesh numbers.

| Case No | Total Mesh No | Mesh No along $x$ | Mesh No along $y$ | Mesh No around the Pipeline | $C_{d,rms}$ | $C_{l,rms}$ |
|---------|---------------|-------------------|-------------------|------------------------------|-------------|-------------|
| Mesh 1 | 132043 | 577 | 80 | 120 | 1.4353 | 0.7945 |
| Mesh 2 | 202288 | 600 | 115 | 160 | 1.4871 | 0.8004 |
| Mesh 3 | 309842 | 750 | 142 | 200 | 1.4540 | 0.8202 |

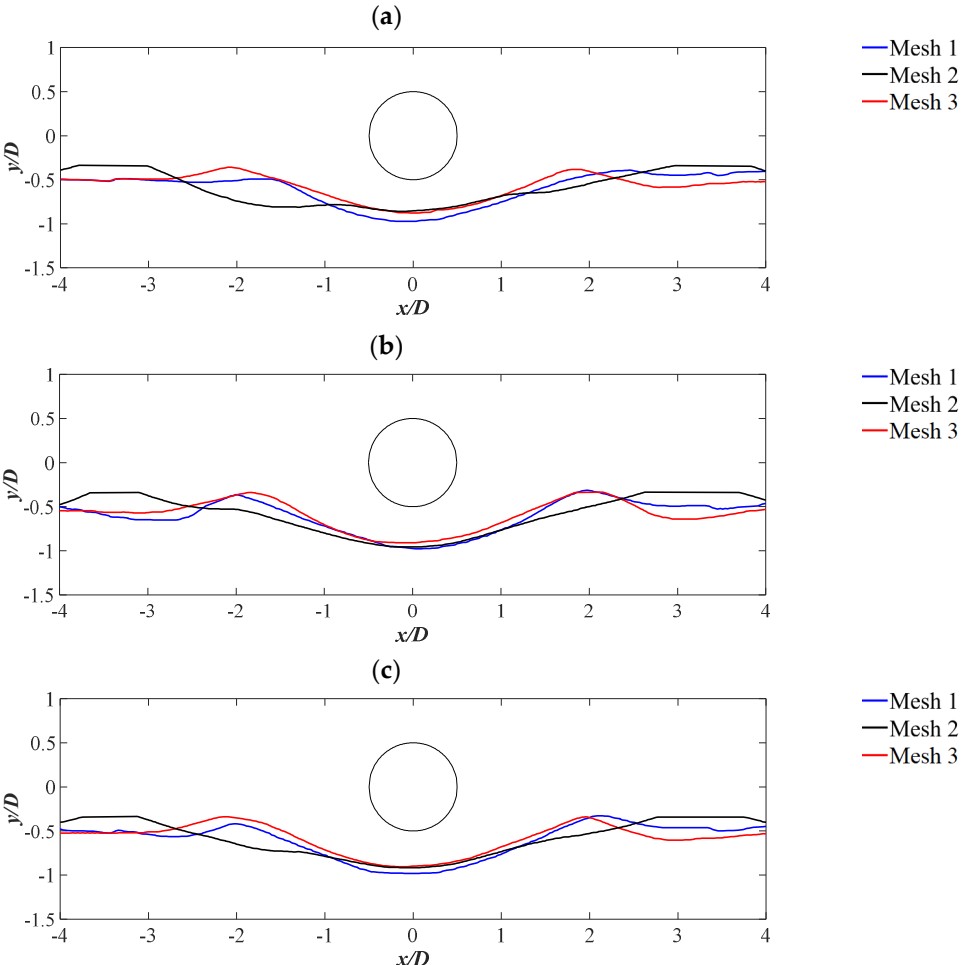

**Figure 2.** The sediment profiles denoted by $\phi = 0.5$ for different meshes: Mesh 1: blue; Mesh 2: black; Mesh 3: red at (**a**) $t = 70$ s; (**b**) $t = 90$ s; (**c**) $t = 110$ s.

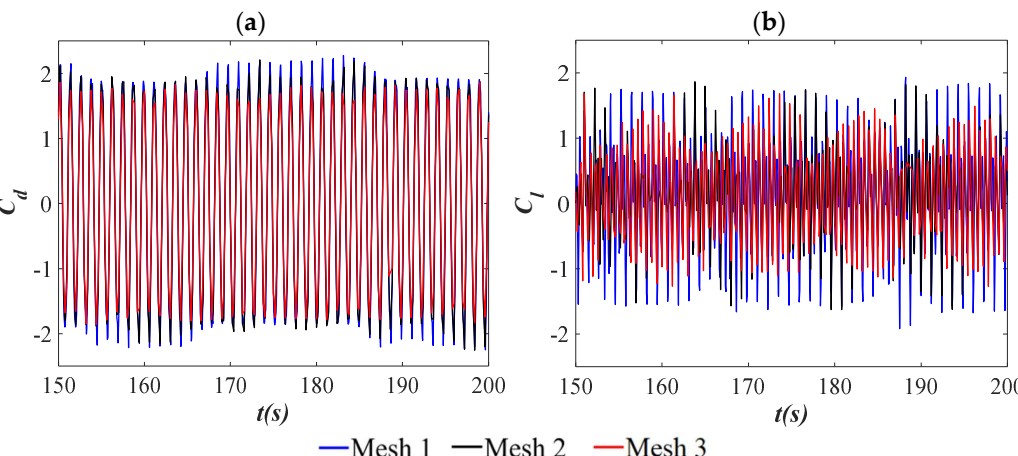

**Figure 3.** The time-histories of (**a**) $C_d$ and (**b**) $C_l$ for different meshes: Mesh 1: blue; Mesh 2: black; Mesh 3: red.

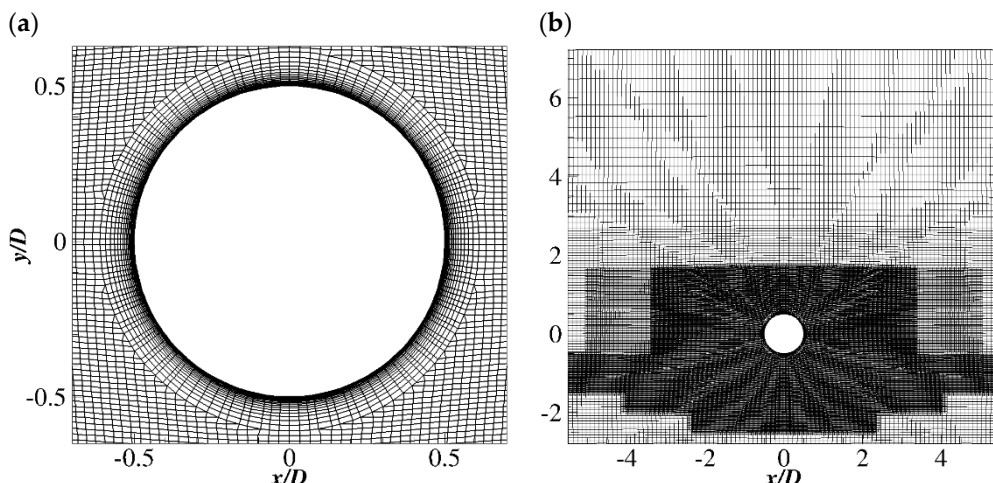

**Figure 4.** An example of Mesh 2 (**a**) around the pipeline and (**b**) around the seabed.

The present two-phase numerical model has been validated for scour beneath a single pipeline under a current flow condition in Yin et al. [30] and Mathieu et al. [4], while in the present study, it is validated for the oscillatory flow condition using Mesh 2 in Table 1. Figure 5 shows the present predicted sediment profiles at three instantaneous time steps and the equilibrium state compared with the experimental data reported by Sumer and Fredsøe [2], and the simulation results of Liang and Cheng [10]. It can be seen that the scour holes of the simulation results are in a satisfactory agreement with the previous published results. However, at the equilibrium state, there are sediment accumulations beside the scour hole close to the pipeline, while for the experimental measured profile, the sediment accumulations are weak and tend to be located far from the pipeline. Despite this, the maximum depth of the scour hole can be well-predicted. Furthermore, the time history of the nondimensional scour depth $S/D$ with the nondimensional time $t^*$, defined as $t^* = t\sqrt{g(s-1)d_{50}^3/D^2}$ according to Fuhrman et al. [3], is shown in Figure 6a. The result obtained by Fuhrman et al. [3] and the straight line corresponding to the equilibrium scour depth of $S/D = 0.1\sqrt{KC}$, proposed by Sumer and Fredsøe [2]. Both the present result and that reported by Fuhrman et al. [3] overpredict the final scour hole. The present result is close to the initial development of the scour depth obtained using the single-phase model by Fuhrman et al. [3]. However, the two-stage scour process observed by Fuhrman et al. [3] for $KC = 11$ does not appear in the present simulation. According to the explanation by Fuhrman et al. [3], the large deviation of the scour depth from the experimental measurement is due to a resonance phenomenon, where two trough regions on both sides of the pipeline form and the sediment profile becomes unstable. The secondary scour in the two-scour process is initiated by the erosion of the exposed crest between the two troughs. In the present study, as seen from Figure 6b at $t^* = 0.9$, although the scour depth in the middle of the scour hole is almost the same as that of Fuhrman et al. [3], the two troughs are not observed, which leads to the difference in the sediment profile around the outer part of the scour hole and flat part of the profile from the result of Fuhrman et al. [3]. Therefore, the instability of the sediment profile will not happen, which seems to be more physically sound based on the experimental measurement reported by Sumer and Fredsøe [2].

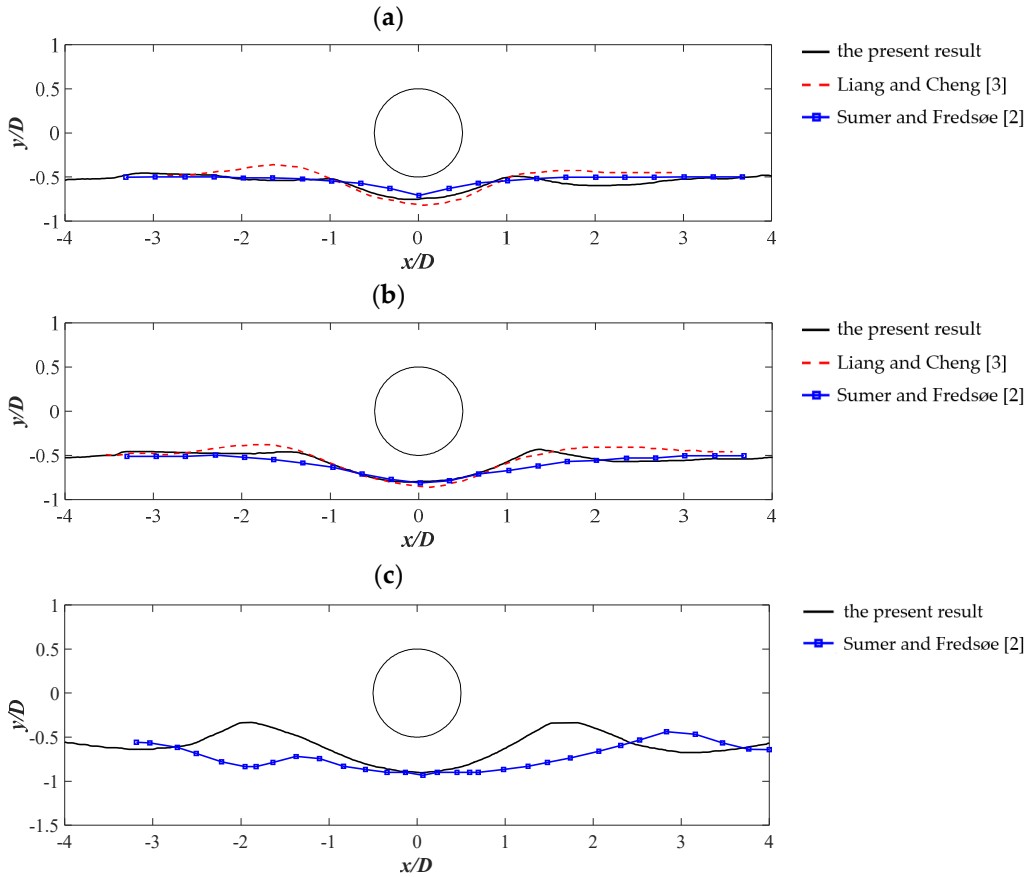

**Figure 5.** Comparison of the present predicted sediment profile obtained using Mesh 2 (solid black line) with the experimental data reported by Sumer and Fredsøe [2] (data adapted from Sumer and Fredsøe, 1990, blue squares) and numerical simulation results obtained by Liang and Cheng [10] (data adapted from Liang and Cheng, 2005, red dashed lines) at (**a**) $t = 30$ s; (**b**) $t = 60$ s; (**c**) $t = 180$ s compared with the equilibrium state of the result by Sumer and Fredsøe [2].

A time-averaged flow field of the fluid phase is shown in Figure 7. The typical steady streaming structures due to the oscillatory flow condition are presented, characterized by two recirculation cells on the top of the pipeline and an outward jet-like flow from the bottom side of the pipeline. The outward jet-like flow leads to the net transport of the sediment, as reported by Fuhrman et al. [3]. The predicted steady streaming structures are similar to those reported in An et al. [31] for $KC = 10$ close to a rigid flat bottom. A zoomed-in figure of the gap beneath the pipeline in Figure 7b shows an inward streaming, both within the sediment and close to the seabed. The inward streaming close to the seabed is due to the induced small recirculation motions, as denoted in Figure 7b, around the sediment profile. This convergent motion has also been observed in An et al. [31] close to the flat bottom and Fuhrman et al. [3] below a reference concentration level of $3.5d_{50}$. However, for the present two-phase flow model, it is indicated that this inward streaming may be also correlated with the fluid flow within the sediment, i.e., seepage flow.

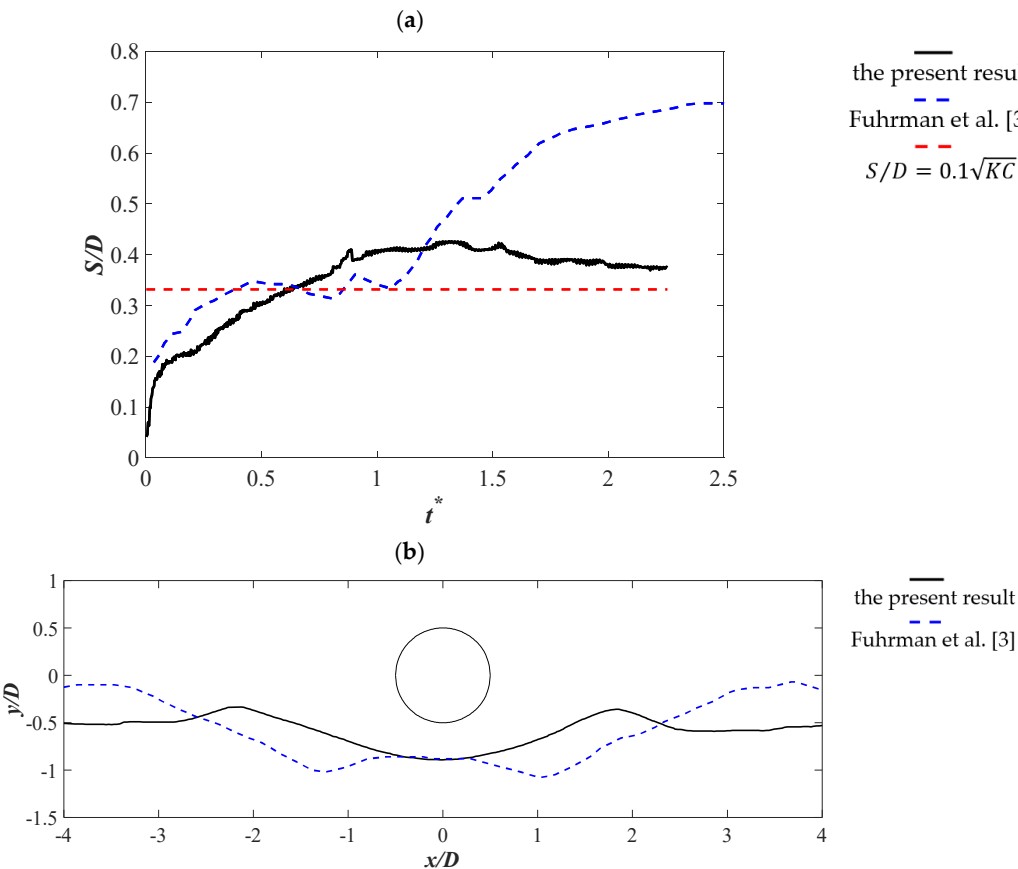

**Figure 6.** (**a**) The time history of the maximum scour depth for the present study (solid black line) compared with that reported by Fuhrman et al. [3] (data adapted from Fuhrman et al., 2014, blue dashed line) and the data predicted by the empirical relationship of $S/D = 0.1\sqrt{KC}$ (red dashed line); (**b**) the present predicted sediment profile (black) compared with that reported by Fuhrman et al. [3] (blue dashed line) at $t^* = 0.9$.

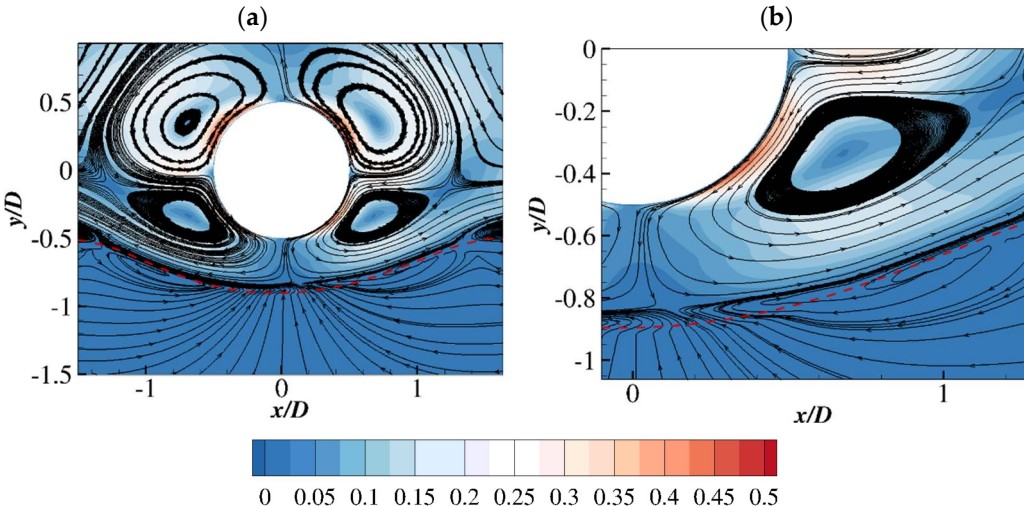

**Figure 7.** (**a**) The time-averaged fluid phase flow streamlines and the contours of the nondimensionalized magnitude of the velocity by $U_m$ for the single pipeline case and (**b**) a zoomed-in figure around the sediment profile. The dashed lines denote the iso-surface identified by $\phi = 0.5$.

## 4. Results and Discussion

### 4.1. Scour Development

The time histories of nondimensional scour depth for the single pipeline and the piggyback pipelines are shown in Figure 8. The total dimensional time of $t = 210$ s corresponding to $t^* \approx 2.3$ are simulated for all cases. It is worth mentioning that due to the sensitivity of the predicted sediment profiles captured using the two-phase model, the equilibrium states of the sediment profiles are dynamic rather than static and there will be variation of the scour depths beyond the simulation time. However, the deviations from the final equilibrium scour depths are not large beyond the simulated time. The horizontal line corresponding to the equilibrium scour depth predicted as $S/D = 0.1\sqrt{KC}$ is also included. It can be seen that for the initial development of the scour process within $t^* \leq 1.5$, the scour depth of the piggyback with $\alpha = 90°$ is larger than those of the other cases. With the decreasing $\alpha$, the scour depth is reduced, which is consistent with the explanation given in Zhao et al. [23] for the steady current condition that the additional small pipeline increases the cross-section area of the configuration, and thus increases the scour depth beneath the pipelines. For $\alpha = 0°$, the initial scour depth is the lowest among all cases. At $t^* > 1.5$, the scour depth for $\alpha = 90°$ is significantly reduced, which may be related to the backfilling process under the oscillatory flow condition. A possible reason for the fast reduction in the scour depth may be further explained by investigating the phase-averaged turbulent kinematic energy (TKE) $k^f$ and resolved $v^{f2}/2K^f$ (for the present RANS model, $K^f$ is defined as $K^f = \left(u^{f2} + v^{f2}\right)/2$) around the pipelines. According to Jang et al. [32], the value of TKE is responsible for the turbulent mixing and the value of $v^{f2}/2K$ can indicate the vertical momentum transfer. Both of the two quantities are important for the suspended sediment transport. Figure 9 shows the contours of the phased-averaged $k^f$ and $v^{f2}/2K^f$ at the phase of $T_w/2$ when the maximum displacement of the oscillatory flow takes place. For the piggyback with $\alpha = 90°$, due to the large cross-section area, a high $k^f$ region is located largely around the piggyback pipeline and a high $v^{f2}/2K^f$ is also restricted within $x/D \leq 1$. Therefore, the sediment transport is largely located around the piggyback pipeline, and after the maximum scour depth is reached, the intensive and localized sediment transport may then lead to the backfilling process. However, for the single pipeline case and the piggyback with $\alpha = 0°$, the high $k^f$ region can spread to $x/D \approx 3$, indicating that the sediment transport happens further away from the pipeline. Therefore, the localized backfilling for the two cases is not obvious.

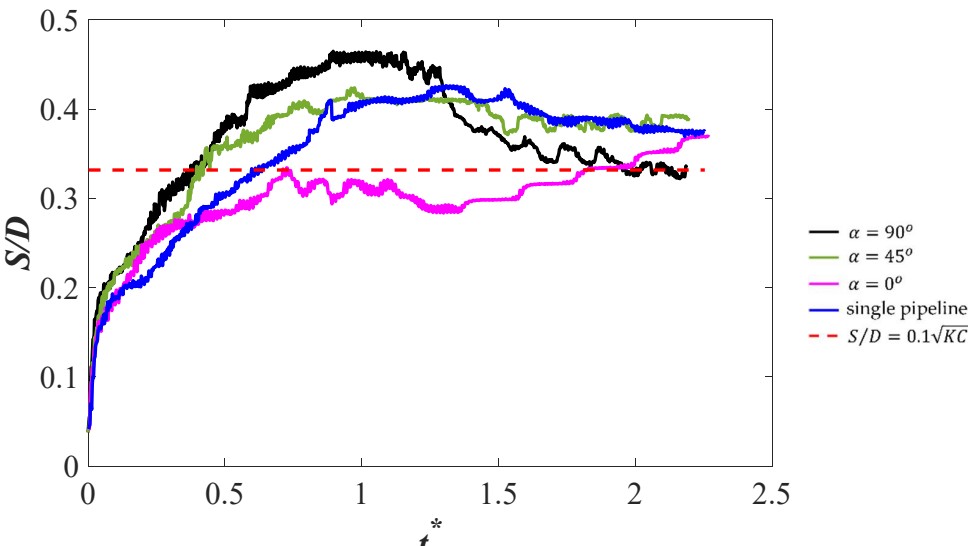

**Figure 8.** The time histories of nondimensional scour depth for the single pipeline and the piggyback pipelines: $\alpha = 90°$: black; $\alpha = 45°$: green; $\alpha = 0°$ magenta; the single pipeline case: blue.

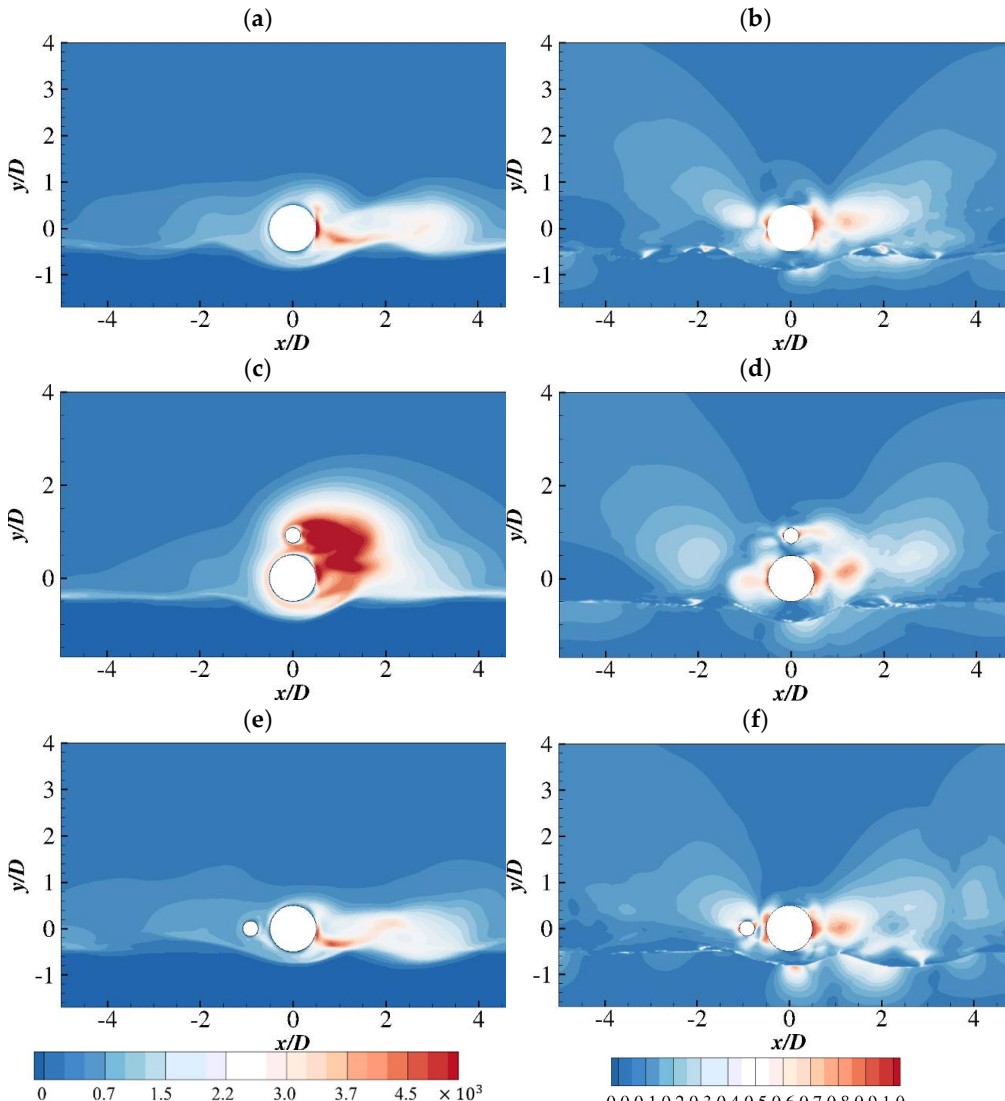

**Figure 9.** (**a,c,e**) The phase-averaged contours of the TKE $k^f$ ($m^2/s^2$) and (**b,d,f**) the value of $v^{f2}/2K^f$ at the phase of $T_w/2$.

It can be observed that the scour depth for the piggyback with $\alpha = 0°$ seems to undergo a slight increase after reaching to an equilibrium state. The continuous increase in the scour depth can be explained by the additional scour hole formed behind the piggyback pipeline. This can be seen in the sediment profiles at three time instants of $t^* = 1$, 1.5 and 2 in Figure 10. The scour beneath the pipeline almost becomes steady, while there is a significant enlargement of the additional scour hole behind the pipeline. The formation of the additional scour hole is consistent with the 'asymmetric shoulders' on either side of the main scour hole observed in Fuhrman et al. [3]. The increasing scour depth of the additional scour hole can be explained by the high $k^f$ region attached to the sediment layer around $x/D \approx 3$ in Figure 9 at the phase of $T_w/2$. In addition, the intensive $v^{f2}/2K^f$ within the sediment layer can also indicate a strong sediment transport around $x/D \approx 3$, which leads to the formation of the additional scour hole. A comparison of the sediment profiles at the three time instants for the three piggyback pipeline cases is also shown in Figure 10. It can be seen that in spite of the additional scour hole behind the pipeline, the scour depth for $\alpha = 0°$ is the minimum among all cases. The small pipeline on the left side of the main pipeline may prevent the formation of the additional shoulders.

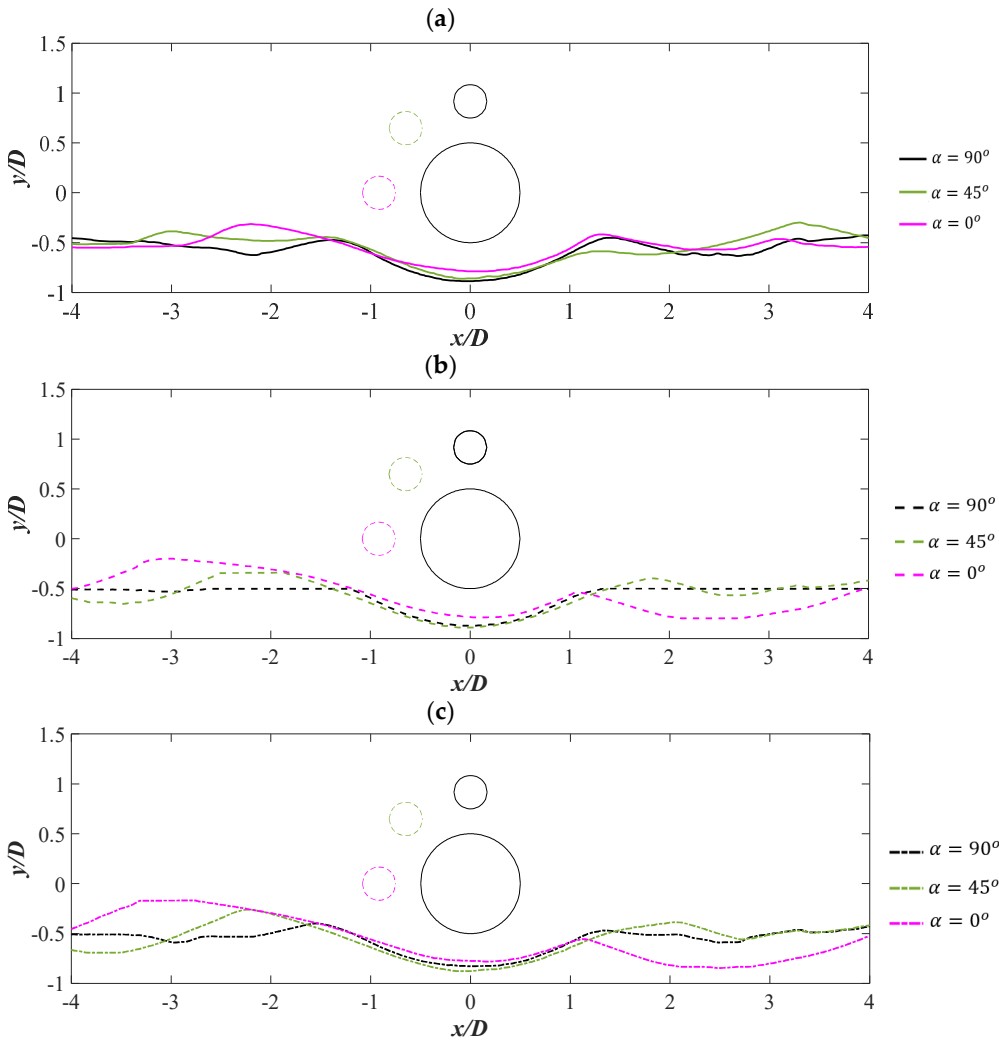

**Figure 10.** The sediment profiles of the piggyback lines with $\alpha = 90°$: black; $\alpha = 45°$: green; $\alpha = 0°$: magenta at (**a**) $t^* = 1$; (**b**) $t^* = 1.5$ and (**c**) $t^* = 2$.

*4.2. Flow Pattern*

The flow patterns for different cases are presented using the phase-averaged velocity field of the fluid phase. The spanwise vorticity and the streamlines for the single pipeline and piggyback pipelines cases are shown in Figures 11–14. For the single pipeline case at the phase of $T_w/3$ in Figure 11b, a recirculation motion begins to be formed, attached to the bottom side of the pipeline rear. There is a positive vortex from the previous cycle washed over the top of the pipeline due to the reverse of the flow. At the phase of $T_w/3$, the separated flow on each side of the pipeline begins rolling up on the top side of the pipeline and grows into a vortex. The wake flow becomes asymmetric. However, due to the relatively small *KC*, the subsequent downstream shedding of the vortex is not observed. Due to the suppression effect of the seabed on the bottom shear layer, only the entrance of the fluid from the top rolling-up shear layer to the bottom shear layer can be seen at the phase of $T_w/2$, and likewise at $T_w$. It is interesting to note that at $T_w/2$ when the oscillatory flow begins to reverse, the fluid flow velocity in the fluid phase domain and within the sediment layer is opposite, and this happens again at the phase of $T_w$, which can be regarded as a unique observation for the present two-phase model. This indicates that the seepage flow may have faster response than the fluid flow domain due to the large pressure within the sediment. The vorticity around the sediment profile is therefore weakened by the two opposite flows at the two sides of the profile. At $T_w/2$, the reverse washing of the formal counter rotating vortex on the top of the pipeline can also be observed.

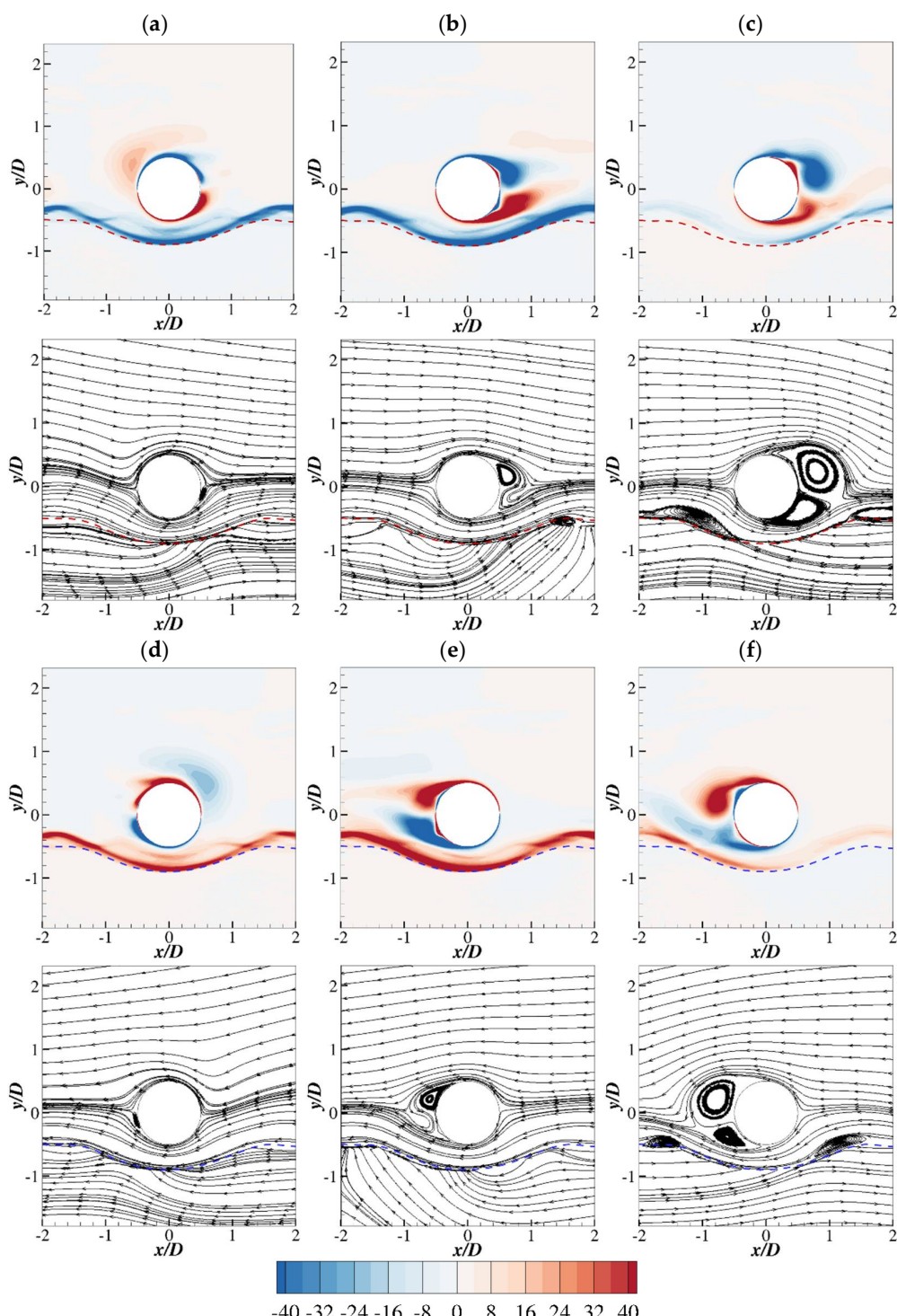

**Figure 11.** The phase-averaged spanwise vorticity and the streamlines for the single pipeline at the phase of (**a**) $T_w/6$; (**b**) $T_w/3$; (**c**) $T_w/2$; (**d**) $4T_w/3$; (**e**) $5T_w/6$; (**f**) $T_w$. The dashed lines denote the iso-surface identified by $\phi = 0.5$.

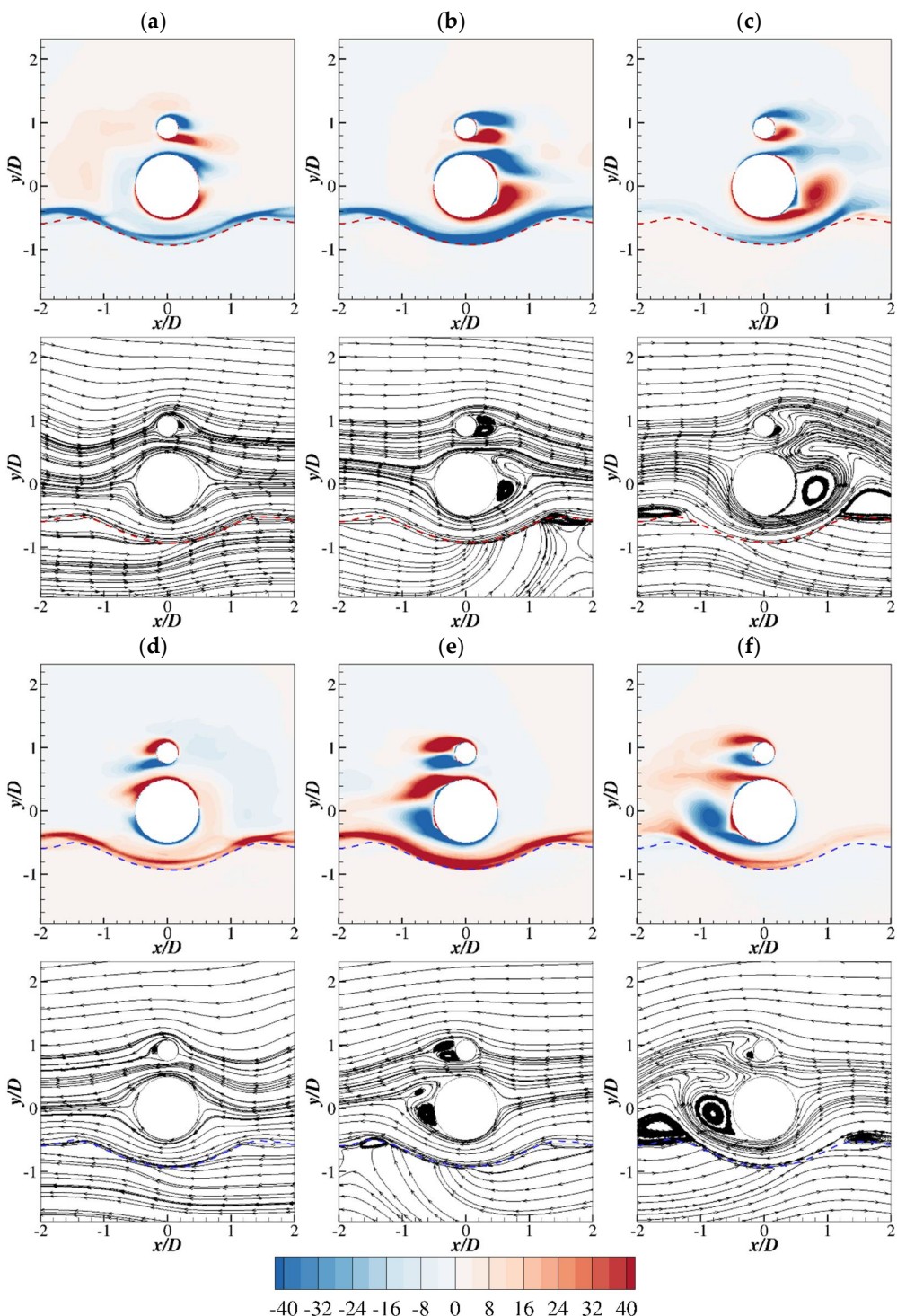

**Figure 12.** The phase-averaged spanwise vorticity and the streamlines for the piggyback pipeline with $\alpha = 90°$ at the phase of (**a**) $T_w/6$; (**b**) $T_w/3$; (**c**) $T_w/2$; (**d**) $4T_w/3$; (**e**) $5T_w/6$; (**f**) $T_w$. The dashed lines denote the iso-surface identified by $\phi = 0.5$.

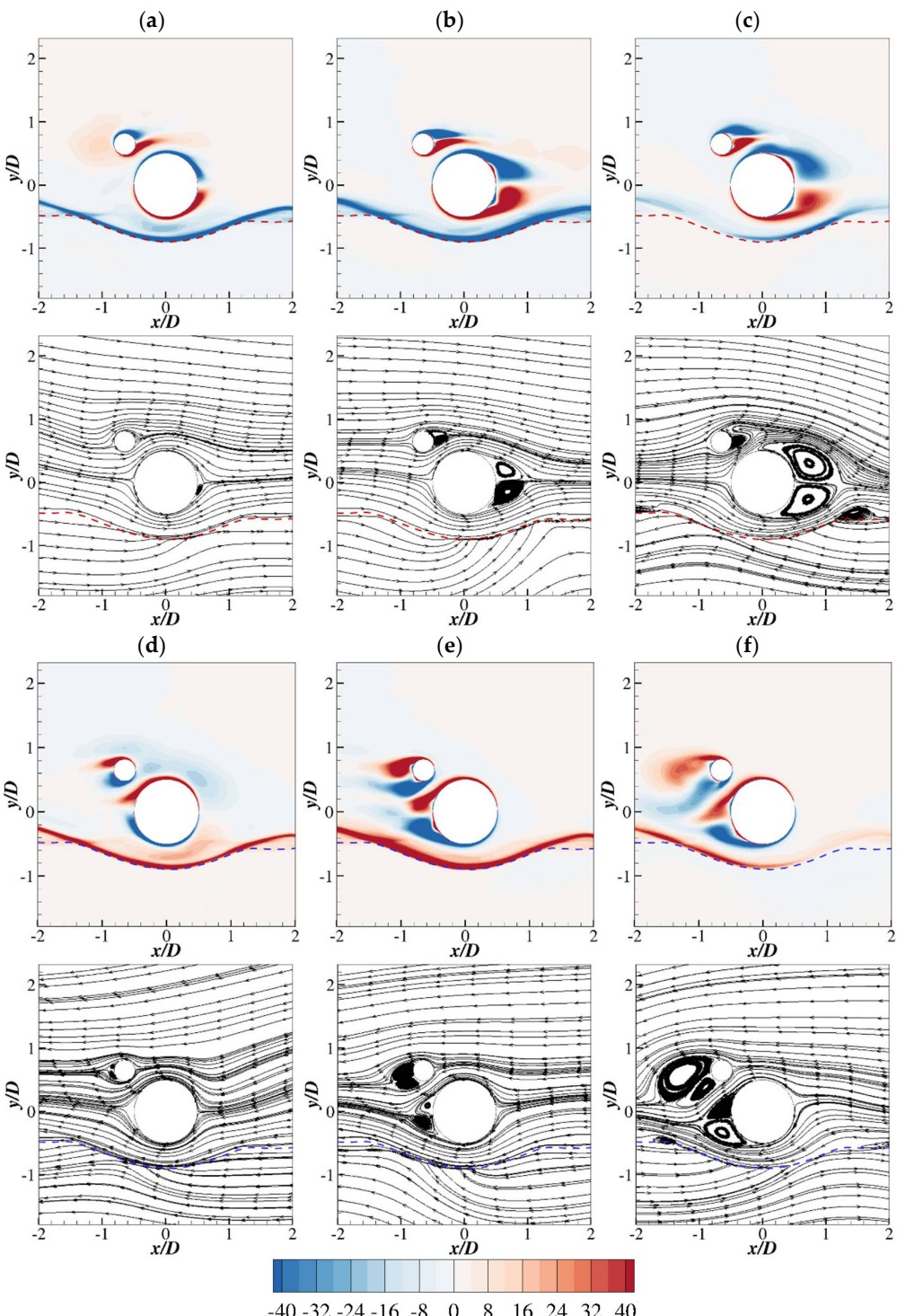

**Figure 13.** The phase-averaged spanwise vorticity and the streamlines for the piggyback pipeline with $\alpha = 45°$ at the phase of (**a**) $T_w/6$; (**b**) $T_w/3$; (**c**) $T_w/2$; (**d**) $4T_w/3$; (**e**) $5T_w/6$; (**f**) $T_w$. The dashed lines denote the iso-surface identified by $\phi = 0.5$.

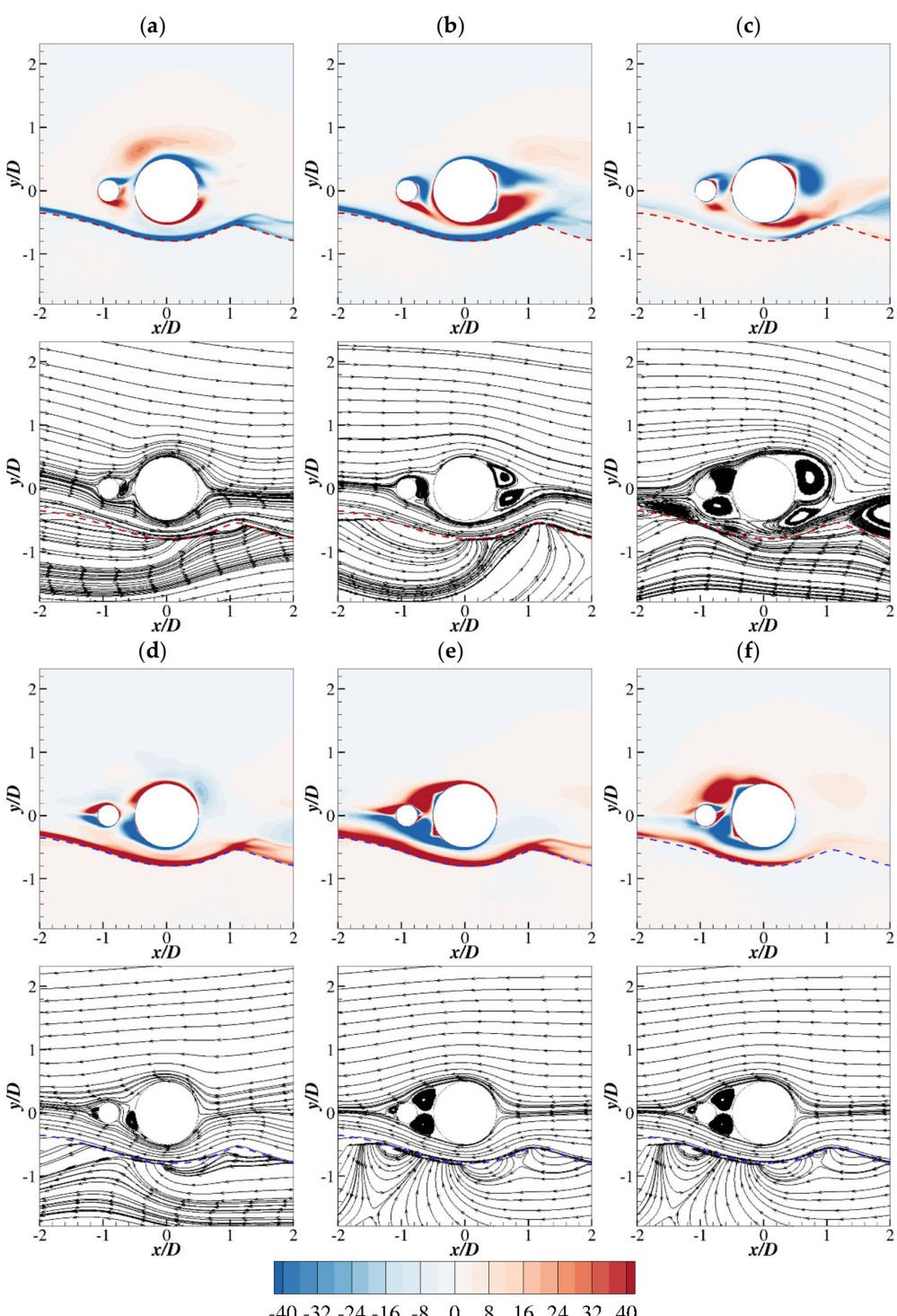

**Figure 14.** The phase-averaged spanwise vorticity and the streamlines for the piggyback pipeline with $\alpha = 0°$ at the phase of (**a**) $T_w/6$; (**b**) $T_w/3$; (**c**) $T_w/2$; (**d**) $4T_w/3$; (**e**) $5T_w/6$; (**f**) $T_w$. The dashed lines denote the iso-surface identified by $\phi = 0.5$.

For the piggyback pipeline cases, the additional smaller pipeline significantly alters the phase-averaged flow fields. At $T_w/6$ and $T_w/3$ for $\alpha = 90°$ and $45°$ in Figures 12 and 13, the interaction of the boundary layers around the two pipelines affects the formation of the rolling up of the vortex. Especially at $T_w/2$ and $T_w$ for $\alpha = 90°$ in Figure 12c,f, the vortex on the top side of the large pipeline is suppressed due to the jet flow through the gap between the two pipelines, which is different from the single pipeline case. For $\alpha = 45°$ at $T_w/3$

and $T_w/2$ in Figure 13b,c, the wake recirculation flow behind the large cylinder tends to be symmetrical. At $5T_w/6$ and $T_w$ in Figure 13e,f, due to the asymmetric of the configuration, the wake flow behind the pipelines is tilted towards the seabed. The strong vortices about to shed behind the small pipeline become weak and diffused as a result of the flow reversal. For $\alpha = 0°$ at $T_w/3$ and $T_w/2$ in Figure 14b,c, there are also strong vortices in the wakes of the two pipelines. At $T_w$ in Figure 14f, the two vortices formed on the left of the large pipeline induce additional small vortices attached to the small pipeline in the gap between the two pipelines.

## 5. Conclusions

In the present study, the two-phase solver, SedFoam, is used to simulate the scour process beneath a single pipeline and a piggyback pipeline with different configurations subjected to an oscillatory flow with $KC = 11$. Two-dimensional simulations combined with the $k - \omega$ 2006 turbulence model are carried out. The numerical model is validated for a single pipeline case against previously published experimental data and numerical simulation results. Then, the scour process beneath a piggyback pipeline with three different configurations is investigated. The main conclusions can be summarized as follows:

- The predicted sediment profiles of the single pipeline case are generally in good agreement with those of the experimental study conducted by Sumer and Fredsøe [2]. The scour depth beneath the pipeline matches that predicted by Fuhrman et al. [3]. However, the drastic increase in the maximum scour depth observed in Fuhrman et al. [3] due to a resonance phenomenon has not appeared in the present simulations. The present predicted equilibrium scour depth is close to that predicted by the empirical formulation in Sumer and Fredsøe [2]. The typical steady streaming structures around the pipeline due to the oscillatory flow condition can be seen from the time-averaged fluid flow field. An inward streaming towards the center of the scour hole is observed within the sediment and close to the seabed.

- The scour depth during the initial development of the scour hole for the piggyback pipeline with $\alpha = 90°$ is the largest among all the investigated cases. However, the scour hole undergoes a backfilling process, which may be due to the strong localized sediment transport around the piggyback pipeline, as indicated by the TKE of the oscillatory flow. Therefore, the scour depth decreases compared with the single pipeline case. When the small pipeline is placed with $\alpha = 0°$, the scour depth right beneath the main pipeline is the smallest among all cases, while there is a large scour hole behind the main pipeline, which may be due to the spread of the intensive sediment transport region downstream. In engineering practice, the small pipeline is usually installed above the large pipeline to avoid being crushed by the large pipeline and the colliding of the small pipeline with the pipeline stinger. However, based on the current investigated flow cases and pipeline configurations, due to the general trend of a decreasing scour hole beneath the piggyback pipeline with the decreasing $\alpha$, it is recommended that the relative angle between the small pipeline and the large one is smaller than $90°$. Further studies are still required to determine an optimal configuration of the piggyback pipeline under different flow conditions.

- The phase-averaged flow fields reveal that at the present investigated $KC = 11$, no obvious downstream vortex shedding can be observed. At the phase of $T_w/2$ and $T_w$ when the oscillatory flow begins to reverse, the directions of the flow in the fluid domain and the seepage flow within the sediment are opposite. The flow patterns are influenced by the additional small pipeline. For $\alpha = 90°$ and $45°$, the interaction of the boundary layers around the two pipelines suppresses the development of the rolling-up vortex on the top side of the large pipeline. For $\alpha = 0°$, the primary vortices between the two pipelines induce additional small vortices attached to the small pipeline and also around the sediment layer.

**Author Contributions:** Conceptualization, J.H., G.Y. and M.C.O.; Methodology, J.H., G.Y., and M.C.O.; Software, J.H. and G.Y.; Validation, J.H. and G.Y.; Formal analysis, J.H., G.Y. and M.C.O.; Investigation, J.H., G.Y. and M.C.O.; Resources, J.H., M.C.O. and X.J.; Data curation, J.H. and G.Y.; Visualization, J.H. and G.Y.; Supervision, M.C.O., D.M. and X.J.; Project administration, J.H., M.C.O. and X.J.; Funding acquisition, J.H. and M.C.O.; Writing—original draft, J.H. and G.Y.; Writing—review and editing, J.H., G.Y., M.C.O., D.M. and X.J. All authors have read and agree to the published version of the manuscript.

**Funding:** This study is supported with computational resources provided by the Norwegian Meta-center for Computational Science (NOTUR), under Project No: NN9372K and The APC was funded by Department of Engineering Research and Design, CNOOC Research Institute, Beijing, China.

**Institutional Review Board Statement:** Not applicable.

**Informed Consent Statement:** Not applicable.

**Data Availability Statement:** Not applicable.

**Conflicts of Interest:** The authors declare no conflict of interest.

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
