# Peer review of "Numerical Investigation of Scour Beneath Pipelines Subjected to an Oscillatory Flow Condition"

_jmse, doi:10.3390/jmse9101102_

Round 1

Reviewer 1 Report

The paper presents a smart application of the SedFoam software to the case of piggyback pipelines subjected to wave-induced scour. The authors concisely describe the 2D numerical model, which involves a coupling of fluid and sediment phases and the use of a k-omega turbulence model, which seems appropriate for the application. The numerical model is validated convincingly against the results of previous experimental and numerical studies; however, the authors may wish to check/improve the presentation of the results in this section (see comments below).

Overall, the manuscript is well-structured and in general well-written (I have noted a few typos/grammatical errors at the end of this review). The authors present a robust computational framework together with novel results which may have important implications for the design of piggyback pipelines. I have the following comments that should be addressed:

  1. Line 182: “the overall shapes of the scour hole beneath the pipeline are similar for different meshes” - Mesh 1 and 3 give a very similar scour hole but it appears much wider in Mesh 2 (which is then selected for further numerical simulations). The authors should comment on this, especially given the shape of the scour hole is used for validation.
  2. Section 3 – it would be useful to present a plot of the different meshes (or parts of the mesh) to help the reader get an idea of the mesh density that is required for the application.
  3. Figure 2, 3, 4, 5, 7, 9: Please add a legend onto the plots to visually describe the different lines, rather than having to interpret the caption.
  4. Line 242: “which seems to be more physically sound based on the experimental measurement” – this sentence is not very clear; it would help to give a reference for the experimental measurement you are referring to.
  5. Figure 4: “Sumer and Fredsoe [10]” should be [2]. The authors should also check if the lines are labelled correctly in this figure – comparing Fig 6 in ref [8], the blue dashed lines presented here should be the experimental data.
  6. Line 319: The reference to Figure 7 should be to Figure 9?
  7. Figure 9: It would be helpful to draw on additional small pipes to illustrate the different piggyback pipe configurations that were considered.
  8. Line 363: “For the single pipeline case at the phase of” – Please give a figure reference here so the reader knows where to look.
  9. Line 472: “it is recommended that the relative angle between the small pipeline and the large on” – I would suggest to include a disclaimer to the effect that the results apply only to the flow case and pipeline configurations considered; further study is needed to generalise/confirm this finding.

Typos/grammar

Line 192: “It worth” – should read “It is worth”

Line 225: “Despite of this” – should read “Despite this”

Line 229, 234: “Fuhrman et al. (2014)” – should be Fuhrman et al. [10]

Line 271, 272 (and elsewhere): there is no need for a hyphen in “inward”

Line 321: “in consistent” should presumably read “inconsistent”

Line 368: “asymmetry” should be “asymmetric”

Reviewer 2 Report

General Comment

The manuscript presents a numerical study to investigate scour beneath a single pipeline and also piggyback pipelines with different configurations, under an oscillatory flow condition and by using a 2D simulation with SedFoam solver. The proposed approach consider the interaction between fluid-pipe-soil interactions. The mathematical framework of the model is presented, as well as its computational implementation. The model is validated based on previous experimental and numerical studies involving a single pipeline. Then, a parametric study is performed for piggyback pipelines with different configurations, namely for different relative angles between the smaller and main pipelines. From the results of this study, among other conclusions, it is recommended to reduce the relative angle to install the smaller pipeline (with respect to 90º) in order to decrease the scour hole beneath the piggyback pipeline.

The topic of the manuscript is very interesting and important since it deals with vortex-induced vibration of pipelines, which influences the stability and fatigue safety under live service. The consideration of an oscillatory flow condition constitutes an important novelty in this study. The submitted manuscript contributes for a better comprehension of the complex interactions and consequences that can arise between fluid-pipe-soil and could be very useful for engineering practice to assess the stability and safety of submarine pipelines.

I made few comments in order to improve the manuscript and also asked for one clarification. The authors should take the few comments into account and revise their manuscript.

Specific Comment 1

Abstract

In the abstract, please avoid to use not defined symbology. Also, add a summary of the main findings of this study

Specific Comment 2

Revise carefully the entire manuscript to correct some typos. For instance, at line 46 it should be “FredsoÆe [2]”

Specific Comment 3

At line 54, please define what is “s” in the sentence “… the scour process under waves at different s …”.

Specific Comment 4

At the beginning of Section3, it is stated that “D = 0.03m”. Is this correct? Did you worked with a scale model? If so, why didn´t you worked with more realistic dimensions for the diameter of the pipeline? Please clarify.

Reviewer 3 Report

The reviewer wants to thank the author for the paper presenting a numerical investigation of the sediment under a big pipe combined with a piggyback one.  S\he has some suggestions and questions:

*1) Abstract: Please introduce KC.

*2) Line (L) 33: please add specific references for the numerical simulations like the experimental investigation.

*3) L54: s is not introduced.

*4) L60: please clarify how those literature references are relevant for this specific paper.

*5) L139: the reviewer would use height instead of width.

*6) L145: how large was the domain presented by Liang and Cheng [8]?

*7) L148: is this gap ratio realistic?

*8) L160/equation (11): At the inlet the sediment has the same velocity as the fluid? So there is a constant fluid and sediment flow coming in, correct?

*9) The outlet was not described.

*10) Figure 1 please introduce the coordinate system.

*11) Grid lines would be helpful for the result figures.

*12) Table 1: please clarify which parameter was changed for each mesh.

*13) L218: Three is not case 2 in Table 1

*14) L219: three instead of two?

*15) Figure 5: Please explain the big differences at x/D $\pm$4.

*16) Figure 3 and 6: the subfigures could be arranged side by side for both figures. This would reduce the white space.

*17) Figure 8: please include a colour bar.

*18) Figure 9: the piggyback pipe should be shown in the same colour as the sediment surface. At the moment it is first confusing hence only one variation is shown.

Thank you very much!

Reviewer 4 Report

Minor suggestions to improve this paper are as follows:

1) please, check the symbol "d", it appears to be used as the diameter of the small pipeline and the "sediment diameter"; it is likely that the "sediment diameter" should be also displayed in one of the figures;

2) the problem considered in this paper involves several Reynolds numbers (for the smaller pipeline in terms of phase 1, for the smaller pipeline in terms of phase 2, for the main pipeline for the phase 1 and for the phase 2), it is better to provide number for all of them;

3) Figs 6&8 require the colormap to interpret the contour.

4) please, clarify what is the total dimensional time t simulated.

5) please, mention the residual values used.

6) please, add more information about the phases, specifically, inlet velocity amplitude Um, viscosities, densities, concentration of particles, that is of particular interest for the sediment phase.

7) related to the above, please, explain the simulation scenario in more details, whether the computational domain is considered filled with both phases of a certain geometry before the simulation has started, or with one phase only and at time t = 0 the second phase only enters the domain?

8) in the paper, currently, several figures indicate a clear border between the sediment and the water phases, but what statistical concentration of particles allows to locate this interface?

9) please, comment whether the scour depth has become stable for the considered cases by the end of the simulation, or further changes could be anticipated beyond the simulation time.
